# Chemical Characteristics of Major Inorganic Ions in PM2.5 Based on Year-Long Observations in Guiyang, Southwest China—Implications for Formation Pathways and the Influences of Regional Transport

**Hao Xiao [1], Hua-Yun Xiao [2], Zhong-Yi Zhang [2], Neng-Jian Zheng [2], Qin-kai Li [1] and Xiao-Dong Li [1],***

[1] School of Earth System Science, Institute of Surface-Earth System Science, Tianjin University, Tianjin 300072, China; xiaohao@tju.edu.cn (H.X.); Qinkai_Li@tju.edu.cn (Q.-k.L.)

[2] Jiangxi Province Key Laboratory of the Causes and Control of Atmospheric Pollution, East China University of Technology, Nanchang 330013, China; xiaohuayun@ecut.edu.cn (H.-Y.X.); zhangzhongyi@ecut.edu.cn (Z.-Y.Z.); zhengnengjian@ecut.edu.cn (N.-J.Z.)

* Correspondence: xiaodong.li@tju.edu.cn

**Abstract:** Sulfate, nitrate and ammonium (SNA) are the dominant components of water-soluble ions (WSIs) in PM2.5, which are of great significance for understanding the sources and transformation mechanisms of PM2.5. In this study, daily PM2.5 samples were collected from September 2017 to August 2018 within the Guiyang urban area and the concentrations of the major WSIs in the PM2.5 samples were characterized. The results showed that the average concentration of SNA ($SO_4^{2-}$, $NO_3^-$, $NH_4^+$) was $15.01 \pm 9.35$ μg m$^{-3}$, accounting for 81.05% (48.71–93.76%) of the total WSIs and 45.33% (14.25–82.43%) of the PM2.5 and their possible chemical composition in PM2.5 was $(NH_4)_2SO_4$ and $NH_4NO_3$. The highest SOR (sulfur oxidation ratio) was found in summer, which was mainly due to the higher temperature and $O_3$ concentrations, while the lowest NOR (nitrogen oxidation ratio) found in summer may ascribe to the volatilization of nitrates being accelerated at higher temperature. Furthermore, the nitrate formation was more obvious in $NH_4^+$-rich environments so reducing $NH_3$ emissions could effectively control the formation of nitrate. The results of the trajectory cluster analysis suggested that air pollutants can be easily enriched over short air mass trajectories from local emission sources, affecting the chemical composition of PM2.5.

**Keywords:** PM2.5; Guiyang; SNA; transformation mechanism; clustering analysis

## 1. Introduction

In recent decades, as a result of increased urbanization and industrialization in China, air pollution, including pollution haze, has become one of the most serious environmental problems impacting the ecosystem, climate change, visibility and human health [1,2]. PM2.5 (particulate matter with an aerodynamic equivalent diameter less than or equal to 2.5 μm) is one of the main factors causing haze and as such, it has received significant research attention [3,4]. PM2.5 are complex fine particles that are, primarily, composed of WSIs (water-soluble ions), carbonaceous species, minerals and metals [5,6]. Sulfate, nitrate and ammonium (SNA) are the major water-soluble ionic species in PM2.5 and have been demonstrated to be associated with the process of haze formation. SNA also form the main acidic and alkaline ions that influence the pH of PM2.5, causing environmental acidification [7,8]. Generally, the generation of SNA depends on the gas-to-particle transformation of their precursors (such as $SO_2$,



$NH_3$ and $NO_x$) under conducive meteorological conditions, following various heterogeneous and homogeneous reaction pathways [9,10].

Recently, studies focused on PM2.5 have been conducted in several cities in China. These studies have mainly described the chemical characteristics of the WSIs in PM2.5 and have attempted to explore the formation mechanisms and sources of PM2.5 [11–13]. The results of these studies have indicated that the chemical composition of PM2.5 differs significantly based on the pollution conditions as a result of variation in the sources or formation processes [14,15]. In particular, the concentrations of SNA have been shown to vary with variation in their precursors or formation processes, which in turn, affects the concentration of PM2.5 [7,16]. In addition, long-distance and short-distance transport of air pollutants have been shown to play key roles in determining the chemical composition of PM2.5 [17,18]. It should be noted that most observations to date have focused on a typical haze episode or samples from a particular season and hence, our understanding of pollution patterns throughout the year is limited. In addition, most studies have focused on large/mega cities, such as the Yangtze River, the Pearl River Delta and the Beijing-Tianjin-Hebei region [3,4,9,11,17,19,20]. However, it has been reported that regional differences and variation in terrain may have a significant influence on aerosol formation and thus, the chemical characteristics and composition of aerosols may vary significantly [10]. For example, Guo et al. [21] reported that variation in topography, climate and emission sources caused changes in the characteristics of WSIs.

Guiyang is an important industrial city in Southwest China housing industries such as rubber, metallurgy, energy extraction and electricity generation, among others [22,23]. The city has a large and dense population (the population of Guiyang exceeded 4.80 million in 2017) and has experienced a rapid increase in the level of traffic (the number of cars exceeded 0.9 million). Guiyang suffers severe acid deposition due to the coal combustion in Southwest China over the past few decades [24]; on the other hand, Guiyang is also an important tourist resort. In recent years, with the development of tourism and economic growth in Guiyang, the negative effects of human activities on ecological environment have intensified [25] and there were some studies on air quality in the city [22–24]. For example, Liang et al. [22] has found that the industrial emissions, automobile exhaust and waste incineration were the main factors affecting PM2.5 in autumn. Xiao et al. [23] has reported that SNA were anthropogenic species, which depend on the coal consumption in Guiyang. However, most of the aforementioned studies only focused on a typical haze episode or samples from a particular season. Therefore, based on long-term and continuous observations, the comprehensive measurement of the chemical characteristics of PM2.5 in Guiyang will be crucial.

This study aims to better understand the chemical characteristics of the major ions in the PM2.5, based on continuous daily observations over a one-year period in Guiyang, Southwest China. Here, we discuss the characteristics of PM2.5, its major ions and their seasonal variation. This study also explores the formation mechanisms of $SO_4^{2-}$ and $NO_3^-$ from their gaseous precursors ($SO_2$, $NO_2$) under different meteorological conditions. In addition, we conducted a backward trajectory cluster analysis to examine the impact of regional transport on PM2.5.

## 2. Materials and Methodology

### 2.1. Description of the Location and Sample Collection

Guiyang city, the capital of Guizhou Province, is located in Southwestern China (E 106°07′ to E 107°17,′ N 26°11′ to N 27°22′) and is surrounded by mountains [24]. The city is covered with typical karst landforms and has a subtropical monsoon climate. The annual average temperature is 15.49 ± 7.51 °C, the wind speed is 2.50 ± 0.66 m s$^{-1}$, the relative humidity (RH) is about 77.72 ± 12.29% and the total precipitation was 1189.2 mm during the sampling period (Table 1) (http://www.weatherandclimate.info/).

**Table 1.** The Seasonal and annual concentration of water-soluble ions and PM2.5 and meteorological parameters during sampling in Guiyang (Mean ± SD, except the precipitation) and the precipitation was the total values during each season.

| Species | Autumn | Winter | Spring | Summer | Annual Average |
|---|---|---|---|---|---|
| PM2.5 ($\mu g\ m^{-3}$) | 27.83 ± 19.52 | 45.81 ± 20.49 | 37.41 ± 18.03 | 20.95 ± 10.86 | 33.11 ± 20.00 |
| WSIs ($\mu g\ m^{-3}$) | 17.92 ± 10.79 | 26.26 ± 10.94 | 18.61 ± 8.34 | 11.19 ± 4.98 | 18.52 ± 10.51 |
| $SO_4^{2-}$ ($\mu g\ m^{-3}$) | 9.07 ± 4.94 | 10.33 ± 4.39 | 9.23 ± 4.20 | 5.81 ± 3.09 | 8.62 ± 4.52 |
| $NO_3^-$ ($\mu g\ m^{-3}$) | 2.90 ± 3.60 | 5.92 ± 4.24 | 2.25 ± 1.99 | 0.79 ± 0.46 | 2.97 ± 3.50 |
| $NH_4^+$ ($\mu g\ m^{-3}$) | 3.04 ± 2.18 | 5.24 ± 2.42 | 3.49 ± 2.10 | 1.90 ± 1.17 | 3.42 ± 2.34 |
| $Ca^{2+}$ ($\mu g\ m^{-3}$) | 2.06 ± 0.90 | 3.18 ± 1.66 | 2.90 ± 1.28 | 2.11 ± 0.84 | 2.56 ± 1.30 |
| RH (%) | 79.35 ± 10.20 | 77.65 ± 15.68 | 75.73 ± 12.40 | 78.11 ± 10.03 | 77.72 ± 12.29 |
| T (°C) | 16.34 ± 5.64 | 5.76 ± 4.34 | 17.01 ± 4.37 | 22.95 ± 10.86 | 15.49 ± 7.51 |
| Precipitation (mm) | 131.2 | 84.5 | 454.5 | 519 | - |
| wind speed ($m\ s^{-1}$) | 2.56 ± 0.62 | 2.43 ± 0.58 | 2.68 ± 0.86 | 2.30 ± 0.48 | 2.50 ± 0.66 |

The sampling site was located on Guanshui road (E 106°44′7″, N 26°34′27″) in the Nanming district, Guiyang City (Figure 1). A total of 365 PM2.5 samples were collected on a continuous daily basis from September 2017 to August 2018 using a KC-1000 (Laoshan Electrical Appliance Factory, Qingdao, China) with a PM2.5 impactor. The specific flow rate was 1050 L min$^{-1}$ for 23.5 h and quartz fiber filters (8 × 10 in., Tissuquartz™ Filters, Washington, USA), which were precombusted in a muffle furnace for 240 min at 425 °C, were used to remove impurities such as potential organic matter. All PM2.5 samples were stored in a freezer at −20 °C until chemical analysis.

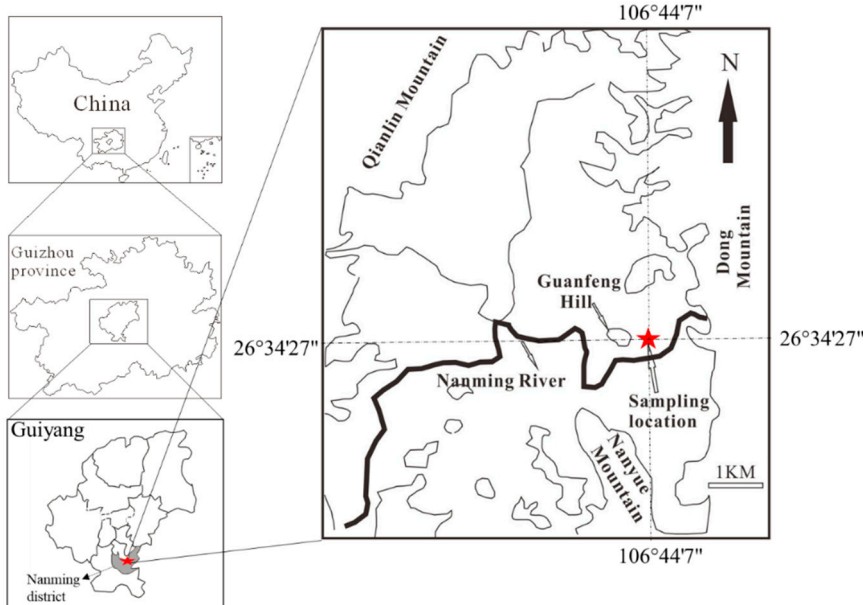

**Figure 1.** Map of sampling sites.

## 2.2. Chemical Analysis

First, 1/8 of each PM2.5 sample filter was cut using ceramic scissors and was placed in a separate clean 50 mL centrifuge tube; 50 mL of ultrapure water was added to each centrifuge tube [26]. Each tube containing a sample was incubated for 30 min under ultrasonication and then shaken for 30 min with a shaker, then, the tubes were centrifuged for 10 min with 2000 revolutions per minute. Finally, the supernatant was filtered using a syringe filter with a 0.22 µm microporous membrane. All filtrate was stored in a refrigerator at −20 °C to determine the concentrations of the inorganic ions. Concentrations of the inorganic ions ($Na^+$, $NH_4^+$, $K^+$, $Mg^{2+}$, $Ca^{2+}$, $F^-$, $Cl^-$, $NO_3^-$ and $SO_4^{2-}$) were measured using ion chromatography (Dionex Aquion (AQ)™, Thermo Scientific™, Massachusetts, USA). The cations ($Ca^{2+}$, $Mg^{2+}$, $Na^+$, $K^+$, $NH_4^+$) were determined using the IC system with a CS12A

4 × 250 mm analytical column and the anions ($SO_4^{2-}$, $NO_3^-$ and $Cl^-$) were determined by the same system with an AS23 4 × 250 mm analytical column.

The extraction efficiency of all ions was calculated by conducting spike-recovery checks. The standard solution of each individual ion was spiked separately to 5 blank filters and analyzed using the same procedure [8]. The extraction efficiency of all ions was observed to be ranging from 85.7 to 109.3%. Three blank samples were also analyzed using the same procedure and the concentrations of the inorganic ions were found to be below the limits of quantification. In addition, the relative standard deviation was less 6% for the reproducibility test.

### 2.3. The Backward Trajectories and Meteorological Data

In this study, two days' (48 h) air mass backward trajectories were calculated using TrajStat software [27] with GDAS (Global Data Assimilation System) data provided by the National Oceanic and Atmospheric Administration (NOAA ARL). A total of 365 backward trajectories of the concentrations of the main ions in the four seasons were used in further analyses. The trajectories over the four seasons were classified into several different clusters by the K-means clustering method [28]. Four clusters were found in the four seasons; the results will be discussed in Section 3.

In addition, the concentrations of gaseous pollutants (such as PM2.5, $SO_2$, $NO_2$ and $O_3$) were obtained from national environmental monitoring stations closest to the sampling site during the sampling period (https://www.aqistudy.cn/).

### 2.4. Statistical Analyses

Statistical analysis was conducted by SPSS 16.0 (SPSS Science, Chicago, USA) and graphs were mainly created with Origin 2018 (OriginLab Corporation, Massachusetts, USA). The *t*-test was used to examine the differences different seasons in this study. Statistically significant difference was set at the level of $p < 0.05$.

## 3. Results and Discussion

### 3.1. Characteristics and Seasonal Variations of PM2.5 and Major Ions

#### 3.1.1. PM2.5

Table 1 shows the average concentration of PM2.5 in Guiyang in each season from 2017 to 2018 in Guiyang. It should be noted that the highest average concentration of PM2.5 was found in winter and the lowest in summer. The same trend was reported in Shanghai and Heze [19,29]. PM2.5 concentration levels were mainly determined by pollution source emissions, air mass migration, gas-to-particle partition and chemical conversion, which are all associated with meteorological conditions [4]. In winter, local residents consume a lot of fuel (including coal) for heating, as the temperature is low (5.76 ± 4.34 °C) in Guiyang; this likely results in high emission of air pollutants (such as $SO_2$ and $NO_x$). In addition, there is significantly less precipitation in winter (84.5 mm) than in summer (519 mm), hence, the scavenging of atmospheric particulate matter is likely weaker in winter than in summer (Table 1 and Figure S1) [30].

#### 3.1.2. Major Inorganic Ions

The annual average total concentration of WSIs was 18.52 ± 10.51 μg m$^{-3}$, comprising 55.93% (17.58–85.32%) of the PM2.5. The difference between the average total concentration of WSIs at each season are significant (*t*-test, $p < 0.05$) and the seasons change trend as follows—summer < autumn < spring < winter. This is consistent with the temporal trend in PM2.5 concentration (Table 1). During the sampling period, the major inorganic ions in PM2.5 included $SO_4^{2-}$, $NO_3^-$, $NH_4^+$ and $Ca^{2+}$, accounting for 94.87% of the total WSIs (Table 1 and Figure 2). It is worth noting that the annual average concentration of SNA ($SO_4^{2-}$, $NO_3^-$, $NH_4^+$) was 15.01 ± 9.35 μg m$^{-3}$, accounting for

81.05% (48.71–93.76%) of the total WSI concentration and 45.33% (14.25–82.43%) of the PM2.5 mass. Furthermore, there were strong correlations between $NH_4^+$ and both $SO_4^{2-}$ and $NO_3^-$ (Figure S2), suggesting that the possible chemical compositions of the SNA in the PM2.5 were $(NH_4)_2SO_4$ and $NH_4NO_3$, which is consistent with previous studies [31]. Interestingly, the concentration of SNA was lowest in summer and highest in winter; the concentrations of $NO_3^-$ and $NH_4^+$ were 2 to 7 times higher in winter than in summer. Such seasonal variation can be attributed to the low temperature in winter, which promotes the formation of particulate $NH_4NO_3$ from $HNO_3$(gas) and $NH_3$(gas), while $NH_4NO_3$ may preferably decompose to gaseous $HNO_3$(gas) and $NH_3$(gas) at high temperatures in summer [32]. The variation in $NO_3^-$ concentration could also be ascribed to the seasonality of $NO_2$ emissions (Figures 3a and 4d) [33]. Compared with other urban studies, the current study found a higher proportion of $Ca^{2+}$ in the PM2.5, which may be related to the typical karst landforms in Guiyang (weathering of rocks) [7,34,35].

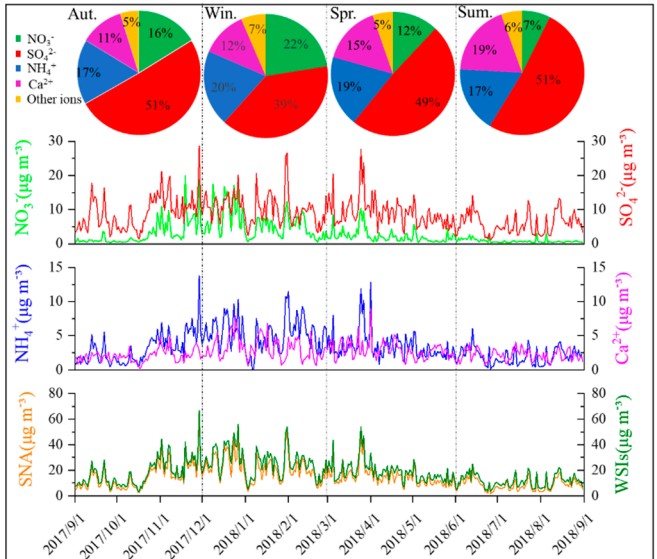

**Figure 2.** Seasonal changes of concentration and proportion of major ions and PM2.5 in Guiyang.

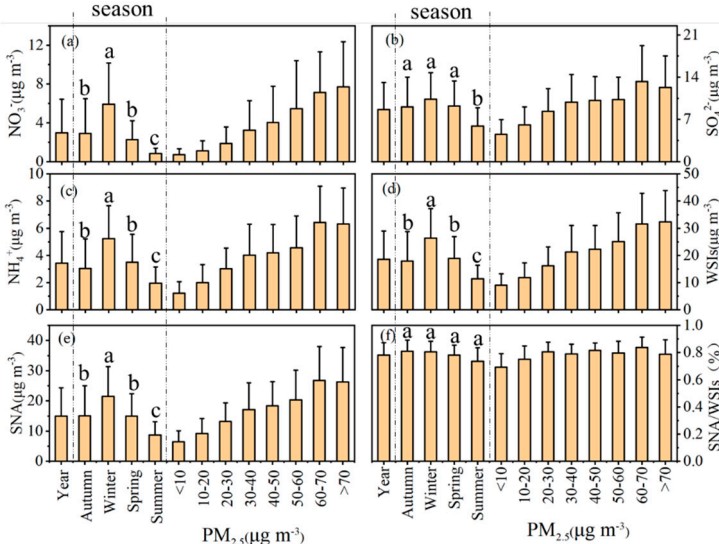

**Figure 3.** Mean ± SD values of (**a**) $NO_3^-$ in PM2.5, (**b**) $SO_4^{2-}$ in PM2.5, (**c**) $NH_4^+$ in PM2.5, (**d**) water-soluble ions (WSIs) in PM2.5, (**e**) Sulfate, nitrate and ammonium (SNA) in PM2.5, (**f**) SNA/WSIs for sampling days, different seasons and the different gradients of PM2.5 concentrations in Guiyang (the different lower cases letters above the boxes indicated that there are significant statistical differences ($p < 0.05$) among seasons).

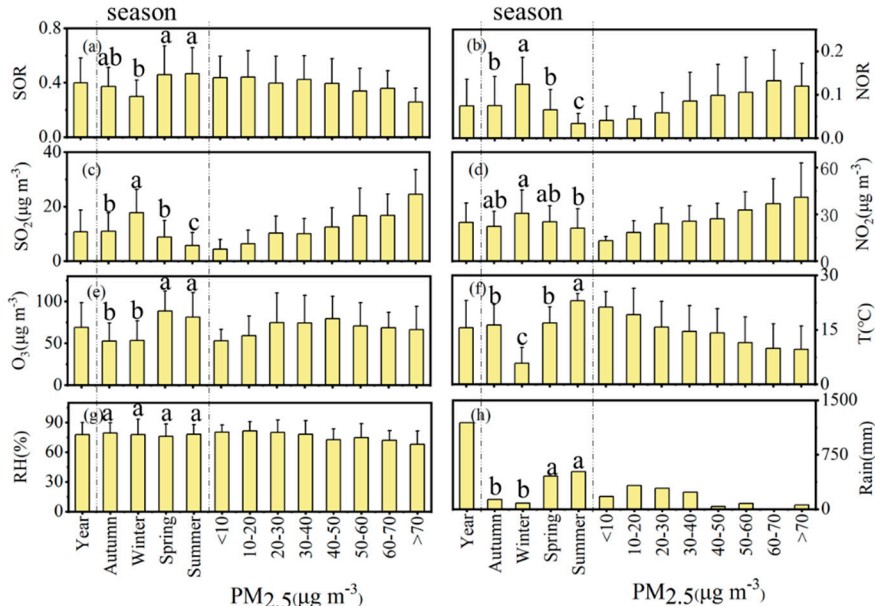

**Figure 4.** Mean ± SD values (except the precipitation) of (**a**) sulfur oxidation ratio (SOR), (**b**) nitrogen oxidation ratio (NOR), (**c**) $SO_2$, (**d**) $NO_2$, (**e**) $O_3$, (**f**) Temperature, (**g**) The relative humidity (RH), (**h**) The total Precipitation for sampling days, different seasons and the different gradients of PM2.5 concentrations in Guiyang (the different lower cases letters above the boxes indicated that there are significant statistical differences ($p < 0.05$) among seasons).

To further explore the characteristics of SNA under different PM2.5 concentrations, the PM2.5 concentrations were divided into eight gradients (<10 µg m$^{-3}$, 10–20 µg m$^{-3}$, 20–30 µg m$^{-3}$, 30–40 µg m$^{-3}$, 40–50 µg m$^{-3}$, 50–60 µg m$^{-3}$, 60–70 µg m$^{-3}$ and >70 µg m$^{-3}$), as shown in Figure 3. It can be seen that the concentration of SNA in PM2.5 increased with the increases in the PM2.5 concentration (Figure 3e). Further, the growth trend of $NO_3^-$ was significantly higher than that of $NH_4^+$ and $SO_4^{2-}$ (Figure 3a–c), indicating that $NO_3^-$ has a greater effect on PM2.5 than $NH_4^+$ and $SO_4^{2-}$; these findings are discussed in more detail in Section 3.2. In addition, the ratio of SNA to WSIs changed only slightly with changes in PM2.5 concentration (Figure 3f), suggesting a stable contribution ratio of anthropogenic sources under different PM2.5 concentrations.

The SNA concentrations of other cities over the world are showed in Table 2. It can be seen that the concentrations of SNA in this study are higher than in Xishan Forest park (7.34 µg m$^{-3}$) and Qinghai-Tibet Plateau (2.06 µg m$^{-3}$), where the influence of human activities is relatively small [21,36]. It indicates that the influence of the contribution of human activities on PM2.5 are significant in Guiyang. However, it is clear that the concentrations of SNA are higher in northern than southern cities in china, such as the Beijing (53.27 µg m$^{-3}$), Tianjin (52.41 µg m$^{-3}$), Shijiazhuang (75.42 µg m$^{-3}$), Hefei (38.53 µg m$^{-3}$) [32,37]. This phenomenon can be attributed to the following two reasons. On the one hand, the economic and population growth may lead to more pollutants were emission. On the other hand, local residents consume a lot of coal for heating under the cold season. Compared with other cities of the world, the concentrations of SNA in Guiyang are close to the Quang Ninh (14.00 µg m$^{-3}$) in Vietnam and Seoul (16.08 µg m$^{-3}$) in Korea and higher than Lowa (4.01 µg m$^{-3}$) in America [38], Fukuoka (8.84 µg m$^{-3}$) in Japan [39] and Masovian (5.78 µg m$^{-3}$) in Poland [40], which showed more serious pollution in Guiyang. Overall, PM2.5 pollution in Guiyang is still exists and it also need taken to the effective control-measures to alleviate.

**Table 2.** SNA ( $\mu$g m$^{-3}$) of PM2.5 in different cities (regions) over China and world.

| | Site | SO$_4{}^{2-}$ | NO$_3{}^-$ | NH$_4{}^+$ | SNA | Time | Reference |
|---|---|---|---|---|---|---|---|
| China | Guiyang | 8.62 | 2.97 | 3.42 | 15.01 | 2017–2018 | This study |
| | Kunming | 6.91 | 2.52 | 2.64 | 12.07 | 2017–2018 | [21] |
| | Nanning | 8.63 | 3.82 | 3.42 | 15.87 | 2017–2018 | [41] |
| | Beijing | 19.44 | 20.32 | 13.51 | 53.27 | 2012–2013 | [37] |
| | Tianjin | 24.26 | 19.62 | 8.53 | 52.41 | 2012–2013 | [37] |
| | Shijiazhuang | 35.66 | 30.47 | 9.29 | 75.42 | 2009–2010 | [37] |
| | Hefei | 15.60 | 15.11 | 7.82 | 38.53 | 2012–2013 | [32] |
| | Tibetan Plateau | 1.43 | 0.41 | 0.22 | 2.06 | 2010–2011 | [36] |
| | Xishan Forest Park | 6.10 | 0.43 | 0.81 | 7.34 | 2013–2014 | [21] |
| Poland | Masovian | 2.17 | 2.44 | 1.17 | 5.78 | 2016 | [40] |
| Japan | Fukuoka | 2.57 | 3.01 | 3.26 | 8.84 | 2015 | [39] |
| America | Lowa | 1.64 | 1.58 | 0.79 | 4.01 | 2011–2012 | [38] |
| India | Pune | 4.80 | 0.89 | 0.51 | 6.20 | 2015–2016 | [42] |
| Vietnam | Quang Ninh | 9.20 | 1.50 | 3.30 | 14.00 | 2009–2010 | [43] |
| Korea | Seoul | 7.39 | 4.75 | 3.94 | 16.08 | 2012–2013 | [44] |

*3.2. Formation Mechanisms of SO$_4{}^{2-}$ and NO$_3{}^-$*

3.2.1. SO$_4{}^{2-}$

As discussed above, SNA accounted for an overwhelming percentage of PM2.5 and played a key role in varying the concentration of PM2.5. Therefore, it is important to explore the SNA formation pathways in order to take measures to control PM2.5 loading. According to previous reports, SO$_4{}^{2-}$ is homogeneously formed through gas-phase oxidation (SO$_2$ + $\cdot$OH$\leftrightarrow$SO$_3{}^{2-}$) and then H$_2$SO$_4$ is formed [10,45]. In addition, aerosol-phase SO$_4{}^{2-}$ can be formed via heterogeneous reactions with H$_2$O$_2$/O$_3$ oxidation of sulfur under metal catalysis and in-cloud processes [45].

Generally, the SOR (sulfur oxidation ratio) is used as an indicator of the extent of the SO$_4{}^{2-}$ formation process [46]. The SOR was calculated as follows:

$$\mathrm{SOR} = \frac{n(SO_4^{2-})}{n(SO_4^{2-}) + n(SO_2)},$$

(1)

where $n$(SO$_4{}^{2-}$) and $n$(SO$_2$) refer to the molar concentrations of SO$_4{}^{2-}$ and SO$_2$, respectively. According to a previous report, when SOR > 0.1, strong photochemical oxidation transforms SO$_2$ to SO$_4{}^{2-}$ [47]. As shown in Figure 4a, the SOR values in this study were higher than 0.1, indicating that significant SO$_2$ oxidation occurred in all four seasons. During the study period, SOR varied seasonally as follows—summer > spring > autumn > winter. The highest SOR values and the lowest SO$_2$ level were observed in summer, indicating stronger transformation of SO$_2$ to SO$_4{}^{2-}$. Meng et al. [48] reported that high solar radiation and higher HO$\cdot$ concentrations are the key factors that promote homogeneous transformation of SO$_2$ to SO$_4{}^{2-}$. In addition, higher O$_3$ concentrations may contribute to the oxidation of SO$_2$ in summer [48,49]. The scavenging effect of precipitation on SO$_2$ may also play a significant role [8]. Meanwhile, highest SO$_4{}^{2-}$, the lowest SOR and highest SO$_2$ levels were found in winter (Figures 3b and 4a,c). It is reasonable to assume that a large amount of SO$_2$ would have been emitted from coal burning and this can significantly accumulate under stagnant weather conditions during winter [50]. Therefore, even in the case of low SOR, higher SO$_2$ may still lead to more SO$_4{}^{2-}$ production in winter. Additionally, lower solar radiation during winter can reduce photochemical activity under stagnant weather conditions and inhibit the oxidation of SO$_2$ to SO$_4{}^{2-}$ [51]. Over all,

the higher temperature and HO· may lead to higher SOR in summer, while the much higher $SO_2$ concentrations and stagnant weather condition will cause more $SO_4^{2-}$ production in winter although the transformation were relatively low.

As shown in Figure 4, the PM2.5 concentrations were divided into eight gradients. We found that the concentrations of $SO_2$ and $SO_4^{2-}$ increased with increasing concentration of PM2.5 (Figures 3b and 4c), suggesting that the loading of $SO_2$ and/or $SO_4^{2-}$ could have been a controlling factor in PM2.5 pollution. As expected, the lowest SOR value and the highest concentration of $SO_2$ were both observed in the highest PM2.5 concentration (Figure 4a,c). This can primarily be attributed to lower solar radiation and higher $SO_2$ emissions under hazy weather.

### 3.2.2. $NO_3^-$

$NO_3^-$ is mainly formed through homogeneous and heterogeneous reactions involving oxidation of $NO_2$ by OH radicals and hydrolysis of $N_2O_5$ [14,52]. The NOR (nitrogen oxidation ratio) is used as an indicator of the extent of the $NO_3^-$ formation process [46]. The NOR was calculated as follows:

$$\text{NOR} = \frac{n(NO_3^-)}{n(NO_3^-) + n(NO_2)},$$

(2)

where $n(NO_3^-)$ and $n(NO_2)$ refer to the molar concentrations of $NO_3^-$ and $NO_2$, respectively. The NOR exhibited the opposite seasonal trend to the SOR—winter > autumn > spring > summer. Generally, higher solar radiation during summer can enhance photochemical activity.

However, as shown in Figure 4b, the NOR and average temperature exhibited opposite trends. This may be due to volatilization of nitrates being accelerated at higher temperatures [32]. Furthermore, the NOR was highest in winter (>0.1), which indicates that more $NO_2$ was oxidized to $NO_3^-$. The Meng et al. [48] has reported that the heterogeneous reactions main occur in winter and the availability of HO· for $NO_2$ oxidization become lower with increasing $NO_2$ concentrations [53]. Meanwhile, relatively higher proportions of $NO_2$ would be oxidized by the $O_3$, even though the $O_3$ concentrations were lower in winter. Additionally, RH can have a strong influence on $SO_4^{2-}$ and $NO_3^-$ formation, especially when RH > 60% [54]. However, in this study, almost no correlation was observed between RH and SOR, nor between RH and NOR. Since the RH in all seasons was above 70% and exhibited consistent variation, it appears that RH is not the primary factor affecting seasonal variation in the SOR and NOR in Guiyang. Similar phenomena have been reported in a previous study [8].

In the current study, the NOR and the concentrations of $NO_3^-$ and $NO_2$ were found to increase with increasing PM2.5 concentration (Figures 3 and 4). The possible reasons are discussed below. Generally, $NO_3^-$ is produced by the oxidation reaction of $O_3$ and OH. As ·OH production decreases on hazy days, the oxidization of $NO_2$ via the $O_3$ pathway might increase with increasing $NO_2$ emissions; this could result in a higher NOR and would have a positive impact on the concentration of PM2.5 [55]. Thus, in our study, $NO_3^-$ formation may be dominated by the $O_3$ oxidation pathway during hazy days [53]. On the other hand, as discussed above, there were inverse relationships between temperature and both the $NO_3^-$ and PM2.5 concentrations and more $NO_2$ was oxidized to $NO_3^-$ in winter resulting in an increased PM2.5 concentration. Therefore, the variation in the NOR with PM2.5 concentration might also be ascribed to the changes in temperature.

Pathak et al. [9] has reported that when $NH_4^+/SO_4^{2-} \geq 1.5$ (molar ratio, $NH_4^+$-rich), $NO_3^-$ forms via a gas-phase reaction—$NH_3 + HNO_3 \leftrightarrow NH_4NO_3$ and when $NH_4^+/SO_4^{2-} < 1.5$ (molar ratio, $NH_4^+$-poor), $NO_3^-$ forms via hydrolysis of $N_2O_5$ in the preexisting aerosols and the formation reaction is as follows—$N_2O_5(aq) + H_2O(aq) \leftrightarrow 2NO_3^-(aq) + 2H^+(aq)$. In the current study, 79.2% of the samples were $NH_4^+$-rich ($NH_4^+/SO_4^{2-} \geq 1.5$); this is consistent with results from Chengdu in 2017 [14]. As shown in Figure 5a, there was a significant correlation ($R^2 = 0.56$) between the $NH_4^+/SO_4^{2-}$ molar ratio and the $NO_3^-/SO_4^{2-}$ molar ratio, suggesting that the formation of $NH_4NO_3$ primarily occurred via a

gas-phase reaction in the $NH_4^+$-rich environment [9]. "Excess ammonium" was defined as the amount of ammonium in excess of that required for $NH_4^+/SO_4^{2-}$=1.5, which was calculated as follows [9,56]:

$$NH_4^+\text{-excess} = (NH_4^+/SO_4^{2-} - 1.5) \times SO_4^{2-}, \tag{3}$$

where $NH_4^+$ and $SO_4^{2-}$ refer to their molar concentrations. As shown in Figure 5b, the correlation between $NO_3^-$ and $NH_4^+$-excess ($R^2 = 0.67$, $p < 0.05$) was significantly stronger in the $NH_4^+$-rich environment than in the $NH_4^+$-poor environment ($R^2 = 0.07$, $p > 0.05$), which indicates that homogeneous gas-phase formation of $NH_4NO_3$ was significant when there was "excess $NH_4^+$." Generally, the slope of regression line between $NH_4^+$-excess and $NO_3^-$ closed to 1, suggesting that $NH_4^+$-excess and $NO_3^-$ could be neutralized [7]. In this study, a shallower slope (k = 0.65) between $NH_4^+$-excess and $NO_3^-$ was found with the $NH_4^+$-rich samples, which indicated that $NH_4^+$ could not be completely neutralized by the $NO_3^-$ and the remaining $NH_4^+$ would affect the basicity of the PM2.5 samples [7].

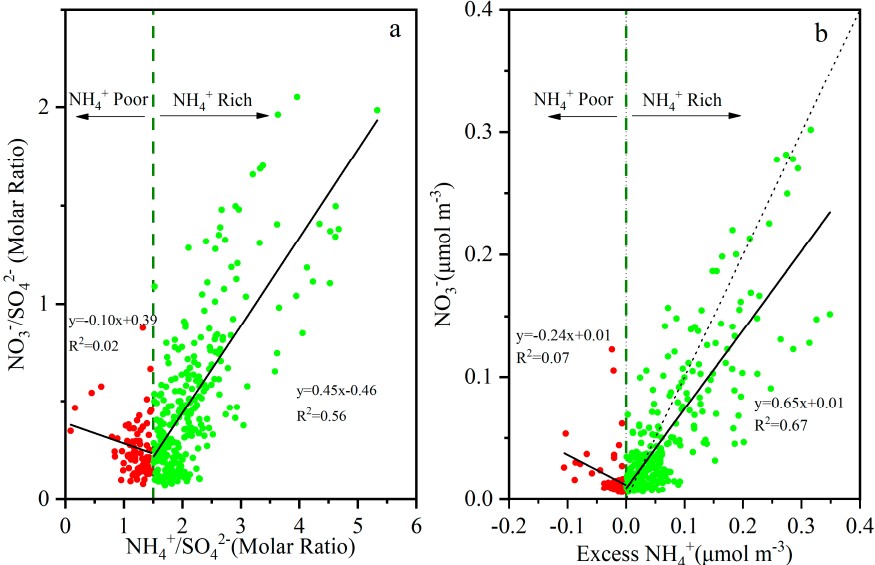

**Figure 5.** Molar ratios (**a**) of $NO_3^-/SO_4^{2-}$ vs. $NH_4^+/SO_4^{2-}$ and the correlations (**b**) of Excess- $NH_4^+$ vs. $NO_3^-$ under different condition ($NH_4^+$−rich or $NH_4^+$−poor).

In addition, a shallower slope (k = 0.65) was found in the $NH_4^+$-rich samples, which indicates that $NH_4^+$ could not be completely neutralized by the $NO_3^-$, the remaining $NH_4^+$ affected the basicity of the PM2.5 samples. However, there was no linear correlation between $NO_3^-$ and the $NH_4^+$-poor samples, indicating that $NO_3^-$ formation was not obvious in the $NH_4^+$-poor environment [7].

In order to further study the effect of $NH_4^+$ on $NO_3^-$ formation, the relationship between $NH_4^+$ and NOR was evaluated. As shown in Figure 6, there was a strong linear relationship between NOR and $NH_4^+$ concentration. NOR increased more quickly in the case of $NH_4^+ > 4 \mu g\ m^{-3}$ (Figure 6), which indicates that the $NH_4^+$ concentration might be an important controlling factor for the transformation of $NO_2$. $NH_4^+$ is a secondary product of atmospheric $NH_3$ and reducing $NH_3$ emissions could effectively control the formation of $NO_3^-$ in Guiyang [57].

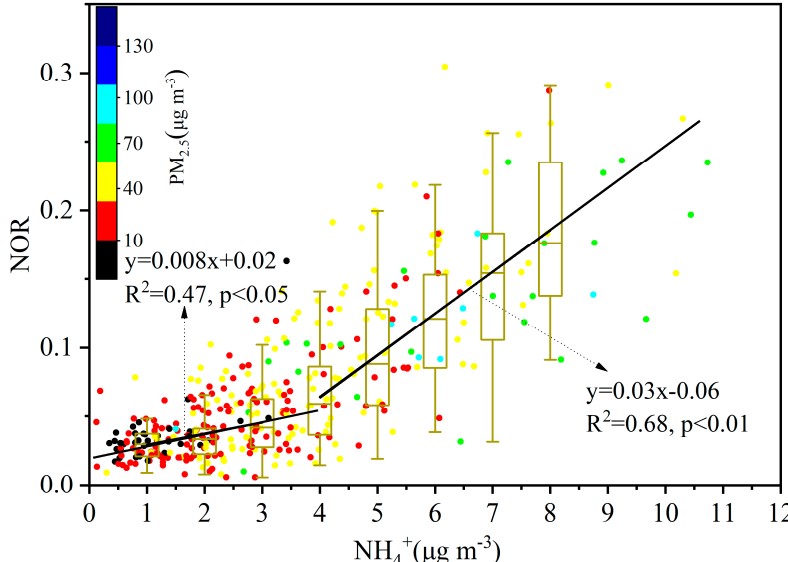

**Figure 6.** Relationship between $NH_4^+$ and NOR and color-coded by PM2.5 concentrations. The NOR data were also binned according to $NH_4^+$ concentrations (<1 µg m$^{-3}$, 1–2 µg m$^{-3}$, 2–3 µg m$^{-3}$, 3–4 µg m$^{-3}$, 4–5 µg m$^{-3}$, 5–6 µg m$^{-3}$, 6–7 µg m$^{-3}$, >7 µg m$^{-3}$), respectively. For each box plot of NOR, the solid line inside the box represents the median, the box encompasses the 25th to 75th percentiles and the bar denotes the 5th and 95th percentiles.

### 3.3. The Clustering Analysis of Backward Trajectory

#### 3.3.1. Trajectory Clustering

Air quality is affected by local emissions and regional transport of air pollutants. Backward trajectories are useful to assess source regions and transport pathways of PM2.5 [8]. In terms of the seasonal changes of the air masses arriving in Guiyang, the major trajectories were divided into four clusters in each season by k-means clustering. The obtained results are shown in Figure 7.

In autumn (Figure 7a), there was no significant difference in the proportion that each of the four clusters of trajectories contributed to the total trajectories. Among the clusters, cluster 2 and cluster 4 originated locally in Guizhou province, accounting for 53.84% of the total trajectories. In addition, cluster 1 (21.98%) and cluster 3 (24.18%) originated from the west of Guangxi province and the north of Hunan province, respectively. In contrast, in the other three seasons, there were significant differences in the proportion that each of the four clusters of trajectories contributed to the total trajectories. In winter (Figure 7b), cluster 1 was derived from Guizhou province in conjunction with the northwest of Guangxi province; it crossed over the southern part of Guizhou province before arriving at Guiyang city. This cluster accounted for the highest proportion of the total trajectories (54.44%). The other clusters originated from the north of Guizhou province, the west of Yunnan province and northwest of Hunan province and accounted for 17.78%, 5.56% and 22.22% of the total trajectories, respectively. In spring (Figure 7c), cluster 1 (51.09%) accounted for the highest proportion of the total trajectories and was derived from the west of Guangxi province and traveled across the south of Guizhou province before arriving at Guiyang city. Cluster 2 (22.83%) and cluster 4 (18.48%) originated from the northwest and northeast parts of Guiyang city, respectively. Cluster 3 (7.61%) accounted for the lowest proportion of the total trajectories, it originated from the southwest of Yunnan province and crossed over the south of Yunnan province before arriving at Guiyang city. In summer (Figure 7d), cluster 1 accounted for the highest proportion of the total trajectories (47.83%); it originated from the central area of Guangxi province. The other clusters originated from the northeast of Guizhou province, the north of Vietnam and the north of Hunan province and accounted for 22.83%, 15.22% and 14.13% of the total trajectories, respectively. Overall, although the air masses that originated from Guangxi Province accounted for a large proportion across all four seasons in this study, short- and long-distance air transport was evident.

Specifically, air masses that were derived from Yunnan province and Hunan province reflected the features of long-distance air transport (cluster 1 and 3 in autumn, cluster 3 and 4 in winter, cluster 3 in spring and cluster 3 and 4 in summer). In contrast, the air masses that originated from Guizhou province or areas surrounding Guangxi province showed the features of short-distance air transport (cluster 2 and 4 in autumn, cluster 1 and 2 in winter, cluster 1, 2 and 4 in spring and cluster 1 and 2 in summer).

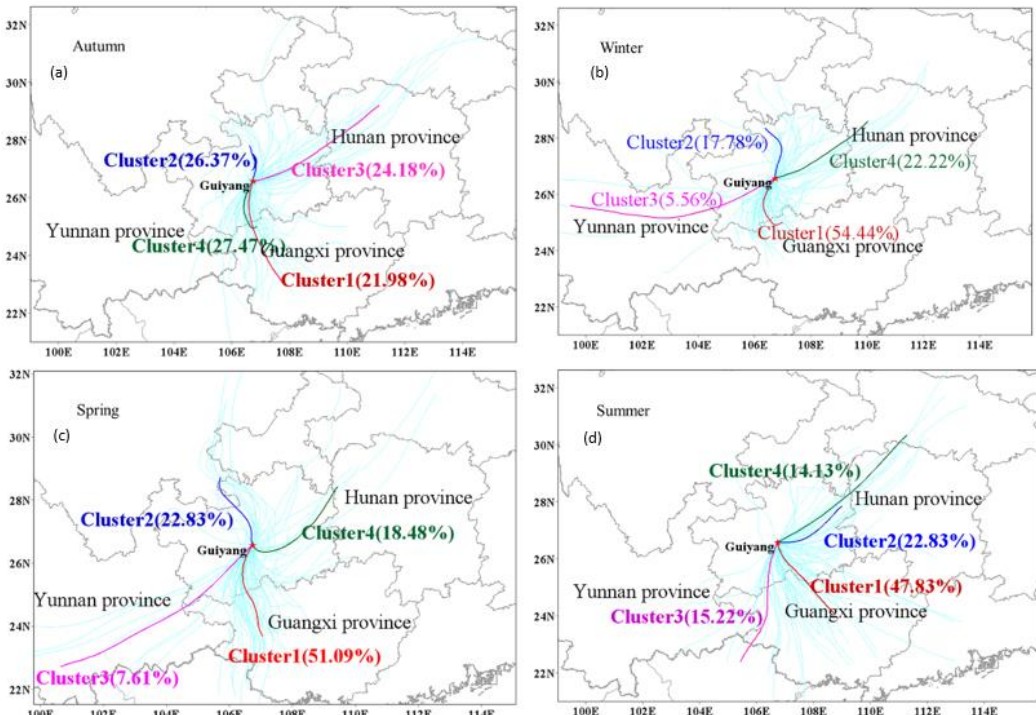

**Figure 7.** The 48 h backward trajectories clustering analysis of air mass arriving in Guiyang. (The thin colored lines indicate that analytical results of the 48 h air mass backward trajectories, thick colored lines indicate that the trajectory clustering results.). (**a**) In Autumn (**b**) in Winter (**c**) in Spring (**d**) in Summer.

### 3.3.2. The Corresponding Concentrations of Species

The corresponding average concentrations of species are shown in Table S1 and Figure 8. The higher concentration of total WSIs was found in short-distance trajectories across the four seasons and accounted for a large proportion of PM2.5, suggesting that the WSIs in PM2.5 are more enriched in short trajectories from local emission sources [58,59]. Interestingly, the higher concentrations of PM2.5 were observed in the longest trajectories (cluster 3) from Yunnan province in winter and spring and the corresponding total WSIs concentration was lower than other trajectories. This could be explained the PM2.5 that carried in the long-distance air mass from Yunnan Province, which could contain a large number of other components. Furthermore, the higher concentration of SNA was observed in short-distance trajectories across the four seasons, which indicated that SNA mainly comes from local sources. In other words, the short-distance trajectories contain greater emissions of primary pollutants, leading to higher concentrations of $NH_4^+$, $NO_3^-$ and $SO_4^{2-}$. On the other hand, lower SNA concentrations were found in long-distance trajectories (cluster 1 in autumn, cluster 3 in winter, cluster 3 in spring and cluster 3 in summer). However, the emissions of gaseous precursors ($NO_2$ and $SO_2$, no data for $NH_3$) in these areas are close to Guiyang, such as Kunming and Nanning [21,41]. Thus, lower SNA concentrations were found in some trajectories, which could be ascribed to these trajectories that reflected long-distance air transport and were fast-moving, so the pollutants may not easily be enriched in these trajectories [58,59]. While some long-distance trajectories also contained a high proportion of SNA. These trajectories originated from Hunan province, which is highly populated and has many

industrial zones and this may result in higher $SO_2$ and $NOx$ emissions [60]. Generally, the differences meteorological conditions may impact on the concentrations of PM2.5, such as the wet scavenging [61]. However, there are no significant effects of the accumulated rainfall to PM2.5 concentration in each backward trajectory in this study (Table S1). Previous studies have shown that precipitation will reduce the concentrations of PM2.5 and the decrease depended more on the precipitation duration than on intensity [61]. Thus, it appears that wet scavenging is not the primary factor affecting the air mass regional transport in Guiyang. It is noteworthy that the highest concentrations of $Ca^{2+}$ were observed in Karst landform areas (Guizhou province, Yunnan province and Guangxi province), indicating that rock weathering is the main source of $Ca^{2+}$, consistent with the results of Section 3.1.

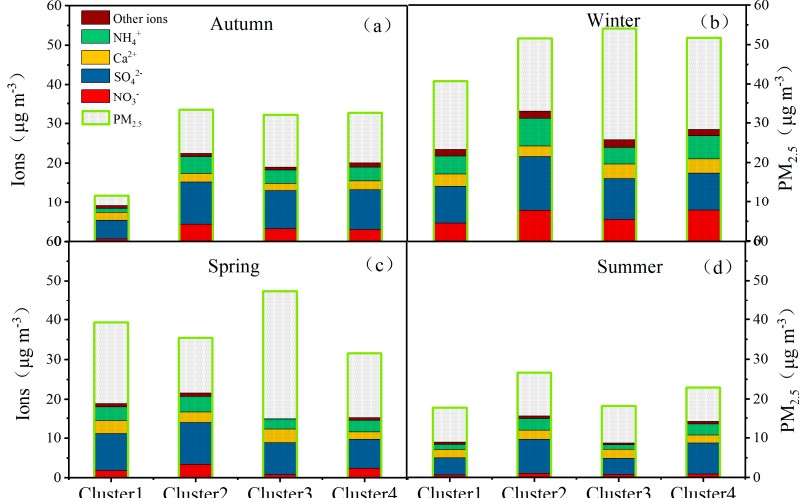

**Figure 8.** Corresponding mean concentration of measured species during sampling period (**a**) autumn; (**b**) winter; (**c**) spring; (**d**) summer).

## 4. Conclusions

In this paper, the total annual average concentration of WSIs accounted for 55.93% (17.58–85.32%) of the PM2.5 mass concentrations. SNA (15.01 ± 9.35 µg m$^{-3}$) as the major WSIs, accounted for 81.05% (48.71–93.76%) of total WSI concentration and 45.33% (14.25–82.43%) of the PM2.5 mass. The results suggest that the PM2.5 was mainly composed of $(NH_4)_2SO_4$ and $NH_4NO_3$. Further, the variations SOR and NOR showed opposite seasonal trends, which were mainly due to meteorological conditions and other factors, such as the temperature and $O_3$ and so forth. This study also found that the NOR showed a similar variation in trend with increasing PM2.5 concentration. This was opposite to the trend of SOR, which implies that the NOR rather than the SOR has a positive effect on PM2.5 concentration. Furthermore, there was a stronger correlation between $NO_3^-$ and $NH_4^+$-excess indicating that the homogeneous gas-phase formation of $NH_4NO_3$ is significant under $NH_4^+$-rich ($NH_4^+/SO_4^{2-} \geq 1.5$) conditions, while $NO_3^-$ formation was not obvious under $NH_4^+$-poor conditions. Further, the NOR was more strongly correlated with $NH_4^+$ concentration, thus reducing the $NH_3$ emissions could effectively control the formation of $NO_3^-$. Trajectory cluster analysis results showed that clusters derived from Yunnan and Hunan reflected the features of long-distance air transport, while air masses that originated from Guizhou province or the juncture of Guizhou province with Guangxi province, showed the features of short-distance air transport. The higher average concentrations of total WSIs and SNA were found in the short-distance trajectories, indicating that they mainly came from local sources and can be easily enriched over short-distances.

**Supplementary Materials:** The following are available online at http://www.mdpi.com/2073-4433/11/8/847/s1, Figure S1: Meteorological parameter and PM$_{2.5}$ concentrations during sampling period, Figure S2: Correlation between $NH_4^+$ and $NO_3^-$, $SO_4^{2-}$, Table S1: The mean concentration of measured species for each type of air mass arriving in Guiyang (Mean±SD, µg m$^{-3}$, Rain as the total rainfall).

**Author Contributions:** Methodology, Z.-Y.Z. and N.-J.Z.; Software, H.X. and Q.-k.L.; Investigation, H.X. and X.-D.L.; Writing-Original Draft Preparation, H.X.; Writing-Review & Editing, H.-Y.X and X.-D.L.; Supervision, X.-D.L.; Project Administration, H.-Y.X. and X.-D.L. All authors have read and agreed to the published version of the manuscript.

**Funding:** This work was supported by the National Natural Science Foundation of China (Grant Nos. 41773006 and 41425014).

**Acknowledgments:** The authors gratefully acknowledge Chandra Mouli Pavuluri and Ding Shi-yuan at Tianjin university for making valuable comments.

**Conflicts of Interest:** The authors declare no conflict of interest.

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
