# Peer review of "Chemical Characteristics of Major Inorganic Ions in PM2.5 Based on Year-Long Observations in Guiyang, Southwest China—Implications for Formation Pathways and the Influences of Regional Transport"

_atmosphere, doi:10.3390/atmos11080847_

Round 1

Reviewer 1 Report

  1. The authors should apply an appropriate statistical method in the interpreting of their data set in the revised manuscript.
  2. Since it is an annual data set analysis, the authors should also discuss the possilble mechanism which cause the differences in seasonal data set variations.
  3. The authors should also make a comparisons of their data obtained in this study to those of the other world regions data of similar related studies in the revised manuscript.
  4. Since the authors are not native English speakers, this manuscript should be aslo edited by an English professor in the revised manuscript. It should include the whole text and grammar.
  5. The authors stated ``Opposite to the SOR, NOR had a more positive effect on PM2.5 concentrations than SOR.'' Why? what the main mechanism leading to this result?
  6. It is an informative study. I recommend this manuscript to be reconsidered for published in this journal after the above major comemnts have been addressed.

Reviewer 2 Report

General comments:

The analysis for this study is a bit shallow. For measuring studies, the researcher would make a lot of assumptions to support their analysis. It is totally fine. However, they have to use persuasive and reasonable theorem to do that. For this study, the reviewer was persuaded by the authors narratives. Please refer the following special comments. There are too many to be revised for the current manuscript to be accepted or published in the future. In conclusion, the reviewer would like to suggest the editor to reject this manuscript and suggest the authors rewrite and resubmit the manuscript.

Special comments:

On line 12&13, you use “are” and “is” to go with the same subject.  The adverbs are not consistent in one sentence.  

On line 21, “Opposite to the SOR, ……than SOR”, please check the narrative again.

On line 22, nitrate is formatted obviously in ammonium-rich environment, so controlling NH3 emission can effectively control nitrate and therefore PM2.5? Are you sure?

On line 62, Guiyang……acid deposition….”.  It is better to denote a reference here.

On line 62, Guiyang is also an important tourist resort, so what is the purpose to mention it?

Table 1, please check the values for precipitation.

Fig. 1, the reviewer could not understand realize Guiyang city is surrounded by mountains in Fig. 1 because there is no indication of terrain heights. Where is the Nanming district in Fig. 1?

Why do you use quartz fiber filters to collect samples? The pre-combustion of quartz fiber filters is to remove impurities of potential organic matters, which is for next OC and EC analysis. However, quartz fiber filter is not suitable for further analysis of WSIs. Instead, Teflon filters is commonly used. Furthermore, you used 1/8 of filters for WSI analysis. How can you be sure that the collected PM2.5 particles are evenly distributed on the filters?

On line 122, please refer Fig. S1 specifically and reasonably. Why do you refer Fig. S1 here?

On line 138, please show the seasonal trend of NOx emissions to support your narratives.

On line 146, the growth trend of nitrate was higher than sulfate and ammonium is not a “new finding” at all nowadays. There are already many studies revealed that. The focus has already changed to deep analysis of formation mechanisms which leads to that phenomenon.

On line 157, the reviewer suggests to use “aerosol-phase” instead of “solid-phase”.

On line 162, what are the previous reports? Also, the SOR is a very old indicator for aging of aerosols. The value 0.1 is a rough value and has limited application. The reviewer suggests the authors not to use this indicator for wide analysis.

Please explain the difference of seasonal trend between sulfate and SOR. The reviewer could not agree the authors’ explanation from line 161 to line 181, which is contradictive to Fig. 3(b).

On line 189, NO2 to NO3- has higher transformation rate at low temperatures?

On line 191, RH strongly influence the formation of sulfate and nitrate. If RH has little relationship with NOR and SOR, what do the authors want to use NOR and SOR to explain the formation of nitrate and sulfate?

On line 197-205, more NO2 was oxidized to NO3- in winter? How to verify that by simply saying NOR was highest in winter? The authors ascribed that to lower temperature in winter. Does it mean the oxidation is stronger under lower temperature environment?

On Line 199-203. Moreover, the authors proposed the assumption of O3 oxidation could dominate the nitrate formation in winter. Does it mean that O3 concentration is higher in winter?

On line 207, “A previous study” indicated the NH4+-rich is highly correlated with gas-phase reaction. Please write out that literature.

When NH4+-poor, did the authors think the NO3- formatted via hydrolysis of N2O5 could not form NH4NO3?

On line 221, the slope 0.65 indicates the NH4+ was remained or NO3- was remained?

On line 227, NOR increased more quickly when NH4+>4 ug m-3,how to judge that via Fig. 6? Could the authors indicate the slopes of NOR for NH4+>4 and < 4 ug m-3?

What is the difference between the scatter plots of NOR via NH4+ and NO3- via NH4+? By Fig. 6, the reviewer could not understand how the NH4+-limited was concluded.

On line 238, please denote the color of each cluster clearly for Fig. 7.

In section 3.3.1 and 3.3.2, the reviewer was totally confused by those cities mentioned. Please add those cities in figures.

On line 267, the highest concentration of total WSIs was found in “cluster 2” across the four seasons? NO, it is not. Please re-analyze and rewrite the manuscript.

Reviewer 3 Report

General comments:

The manuscript “Chemical characteristics of major inorganic ions in PM2.5 based on year-long observations in Guiyang, Southwest China: implications for formation pathways and the influence of regional transport” written by Hao Xiao et al. describes the results of year-long observations of PM2.5 and its chemical compositions and discussed aerosol formation pathways and its seasonal cycles. The topic of the manuscript is within the scope of MDPI Atmosphere. Overall, the manuscript is well written and easy to follow. I would like to consider the publication of the manuscript from MDPI Atmosphere, whereas I have several comments as below, which should be addressed before publication.

Specific comments:

Abstract.

P.1 L.21: What does “positive effect” mean? Does it mean increases in mass concentrations?

  1. Introduction

P.2 L. 58-67: I would suggest that the hypothesis of this study is presented in this paragraph more explicitly. It would help the readers understand what the authors do.

3.1. Characteristics and seasonal variations of PM2.5 and major ions

P.3 L. 118: Do the cold start emissions from vehicles affect the seasonal cycles of PM2.5?

3.2. Formation mechanisms of SO42- and NO3-

P.4 L.171: Why is the sentence “Meanwhile, the lowest SOR and highest SO2 levels were found in winter (Fig. 4a and 4c)” repeated?

P.5 L.191: What oxidation reactions is important for more NO2 oxidation in winter?

P.5 L.220: “a shallower slope”: compared to what?

3.3. The clustering analysis of backward trajectory

P.6, L.274-277: (1) Are there significant differences of meteorological impacts on wet scavenging between the cluster groups? For example, it can be discussed using the accumulated precipitation amounts along each backward trajectory (e.g., Matsui et al., 2011; Ohshima et al., 2012).

(2) Are SO2, NOx, and NH3 emissions higher in Guizhou province than in its surrounding provinces? Spatial maps of a bottom-up emission inventory would help the readers understand the influence of regional transport on the SNA concentrations.

Figures

Figs. 3 and 4: Please clarify what error bars indicate.

Fig. 7: Please clarify what thick and thin colored lines indicate.

References

Matsui, H, Kondo, Y., Moteki, N., Takegawa, N., Sahu, L. K., Zhao, Y., Fuelberg, H. E., Sessions, W. R., Diskin, G., Blake, D. R., Wisthaler, A., and Koike, M.: Seasonal variation of the transport of black carbon aerosol from the Asian continent to the Arctic during the ARCTAS aircraft campaign, J. Geophys. Res., 116, D05202, doi:10.1029/2010JD015067, 2011

Oshima, N., Kondo, Y., Moteki, N., Takegawa, N., Koike, M., Kita, K., Matsui, H., Kajino, M., Nakamura, H., Jung, J. S., and Kim, Y. J.: Wet removal of black carbon in Asian outflow: Aerosol Radiative Forcing in East Asia (A-FORCE) aircraft campaign, J. Geophys. Res., 117, D03204,doi:10.1029/2011JD016552, 2012

Reviewer 4 Report

Xiao et al. analyzed ambient PM2.5 filter samples collected during 2017 - 2018 in Guiyang, China by focusing on water-soluble inorganic ions. The manuscript is overall well written and the results are very well presented. I recommend for publication in Atmosphere after minor revisions.

1. For PM2.5 filter collection, is there a particle size cutter employed at the inlet?

2. Which instrument measures the mass of PM2.5 or where is the data come from? What is the time resolution of PM2.5 mass?

3. What has happened to the remaining part (7/8) of the filter samples?

4. Line 117: in addition to chemical conversion, the gas-to-particle partition is also an important process.

5. Line 170: The sentence is repeated.

6. Fig. 2: I'm just curious if the authors saw huge peaks of some of the ions during the 2018 Chinese new year holiday (Feb 13-20, 2018).

7. Fig. 4h is not a mean value plot.

Round 2

Reviewer 1 Report

  1. QA/QC analysis procedure of the inorganic Concentrations of the inorganic ions (Na+, NH4+, K+, Mg2+, Ca2+, F-, Cl−, NO3−, and SO42−) should be discussed in detail in the revised manuscript. Recovery efficiency for all the inorganic ionic specise should be discussed.
  2. I suggest the authors should update the related ambient air ionic species study in the reference part to the year of 2020 in the revised manuscript.
  3. Table 2 SNA of PM2.5 of various areas study should also update to at least the past 10 years. For example, America area's data is in the years of 2002~2003. Switzerland's data is in the year of 2000 ~ 2003. In addintion, i suggest the Asia area data should include the areas of Korea, India, Vietnam and Mayalysia in the revised manuscript.
  4. This manuscript should be ask for a native English professor to help edit in the revised manuscript. It should be include the whole text and the grrammar.
  5. This manuscript can be accepted for published in this journal after the above minor comments have been addressed.

Reviewer 2 Report

General comments:

It brought much trouble for the reviewer to proceed this review since there is no one-to-one comment-to-reply table. It took time to find where and what was revised. Anyway, the reviewer still agreed to the editor’s request to review this manuscript again. Although many comments were not well-responded, some have been revised according to the reviewer’s suggestion. Therefore, the reviewer suggests the authors to make major revisions for the revision of this manuscript instead of rejection. However, if there is no one-to-one reply table or many comments are still ignored, the reviewer would refuse the next review.

Special comments:

On line 21-23, “SOR (sulfur oxidation ratio) and NOR ……., and O3, etc”. In abstract, the results should be vivid and exact instead of being general or common.

Table 1, what is the unit of time-averaged for precipitation? If the authors did not denote it, how the reader understand the meaning of the values?

Fig. 1, the reviewer still could not catch the location of cities mentioned in the manuscript.

Please explain why do you use quartz filters instead of Teflon filters. What have you done to prove using quartz does not affect the precision of mass weighting and chemical analysis.

On line 133, the reviewer could not find any precipitation information in Fig. S1.

On line 152, the reviewer still could not find the seasonality of NO2 emissions in Fig. 3a and Fig. 4d.

The reviewer is still not persuaded by explanation for the trend of SOR, SO2, and sulfate. The narratives contain many confusions whether sulfate are easily produced or inhibited.

The lower conc. Of NH4+ in summer doesn’t support low NH3 in the same season. Please reconsider the explanation on line 230.

On line 234, the authors mentioned the availability of OH for NO2 oxidation become lower in winter?

On line 234-236, what exactly the O3 pathway is beneficial for NO2 oxidation more in summer or winter?

On line 252, what exactly is the “previous study”?

The review still could not tell why NOR increased more quickly when NH4+>4 ug m-3 via Fig. 6.

On line 268, the authors said NH4+ could not be completely neutralized by the NO3-; however, they said Guiyang is NH3-limited on line 238. Which conclusion is correct?

Again, section 3.3, the reviewer totally could not understand the geography of these cities mentioned in the narrative.

Round 3

Reviewer 2 Report

General comments:

The reviewer would like to apology that he missed the reply file in the last review. The journal review is a good platform for the communication of academic research. After full communications, unreliable or confused knowledge would be reduced for a to-be-published manuscript. The reviewer could understand the authors have tried their best to improve this manuscript. The comments have been largely cut. Please rethink the remaining comments and modify the narratives to avoid confusion and unreasonable explanation. In summary, the reviewer think the manuscript could be accepted and maybe make minor revisions after rethinking the comments below.

Special comments:

On line 152, how to verify the variation of NO3- (Fig. 3a)/NO2(Fig. 4d) concentrations could be ascribed to NO2 emissions? Meteorology could be the main reason as well.

For Fig. 3, what is the difference between a, b, and c, which are the lower case letters above the boxes?

Since SOR is an aging indicator, the higher SOR in summer implies the particles in summer are more aging than in winter, in terms of sulfate formation. However, the SO2 emissions are quite different in summer and winter, therefore, the narratives of SOR would cause confusion and lack of consistent explanation. That’s why the reviewer does not think SOR/NOR are suitable indicators for the explanation of formation mechanism.

In terms of NH4NO3 formation, if Guiyang is a NH4+-rich environment, why reducing NH3 can control the PM2.5? Oppositely, If Guiyang is NO3- limited, isn’t controlling NO3- formation or NO2 emission the right way to reduce PM2.5?
